# Understanding Global Feature Contributions With Additive Importance Measures

**Ian C. Covert**
University of Washington
Seattle, WA
icovert@uw.edu

**Scott Lundberg**
Microsoft Research
Redmond, WA
scott.lundberg@microsoft.com

**Su-In Lee**
University of Washington
Seattle, WA
suinlee@uw.edu

## Abstract

Understanding the inner workings of complex machine learning models is a long-standing problem and most recent research has focused on *local* interpretability. To assess the role of individual input features in a *global* sense, we explore the perspective of defining feature importance through the predictive power associated with each feature. We introduce two notions of predictive power (model-based and universal) and formalize this approach with a framework of additive importance measures, which unifies numerous methods in the literature. We then propose SAGE, a model-agnostic method that quantifies predictive power while accounting for feature interactions. Our experiments show that SAGE can be calculated efficiently and that it assigns more accurate importance values than other methods.

## 1 Introduction

Our limited understanding of the inner workings of complex models is a long-standing problem that impedes machine learning adoption in many domains. Most recent research has addressed this by focusing on *local* interpretability, which explains a model's individual predictions (e.g., the role of each feature in a patient's diagnosis) [25, 30, 34, 38]. However, in some cases users require knowledge of a feature's *global* importance to understand its role across an entire dataset.

In this work we seek to understand how much models rely on each feature overall, which is often referred to as the problem of *global feature importance*. The problem is open to many interpretations [5, 16, 22, 24, 27], and we present the idea of defining feature importance as the amount of predictive power that a feature contributes. This raises the challenge of handling feature interactions, because features contribute different amounts of information when introduced in isolation versus into a larger set of features. We aim to provide a solution that accounts for these complex interactions.

We begin by presenting the framework of *additive importance measures*, a view that unifies numerous methods that define feature importance in terms of predictive power (Section 2). We then present a new tool for calculating feature importance, SAGE,[1] a model-agnostic approach to summarizing a model's dependence on each feature while accounting for complex interactions (Section 3). Our work makes the following contributions:

1. We derive SAGE by applying the Shapley value to a function that represents the predictive power contained in subsets of features. Many desirable properties result from this approach, including invariance to invertible feature transformations and a relationship with SHAP [25].

2. We introduce a framework of additive importance measures that lets us unify many existing methods in the literature. We show that these methods all define feature importance in terms of predictive power, but that only SAGE does so while properly accounting for feature interactions.

3. To manage tractability challenges with SAGE, we propose an efficient sampling-based approximation that is significantly faster than a naive calculation via local SHAP values. Our approach also provides uncertainty estimates and permits automatic convergence detection.

Our experiments compare SAGE to several baselines and demonstrate that SAGE's feature importance values are more representative of the predictive power associated with each feature. We also show that when a model's performance is unexpectedly poor, SAGE can help identify corrupted features.

## 2 Unifying Global Feature Importance Methods

We introduce two notions of predictive power for subsets of features and then discuss our unifying framework of additive importance measures.

### 2.1 Predictive Power of Feature Subsets

Consider a supervised learning task where a model $f$ is used to predict the response variable $Y$ given an input $X$, where $X$ consists of individual features $(X_1, X_2, \ldots, X_d)$. We use uppercase symbols (e.g., $X$) to denote random variables and lowercase symbols (e.g., $x$) to denote values.

The notion of *feature importance* is open to different interpretations, but we take the perspective that a feature's importance should correspond to how much predictive power it provides to the model. Although $f$ is trained using all the features, we can examine its performance when it is given access to subsets of features $X_S \equiv \{X_i \mid i \in S\}$ for different $S \subseteq D$, where $D \equiv \{1, \ldots, d\}$. We can then define "important" features as those whose absence degrades $f$'s performance. As a convention for evaluating $f$ when deprived of the features $\bar{S} \equiv D \setminus S$, we define the *restricted model* $f_S$ as

$$f_S(x_S) = \mathbb{E}\big[f(X) \mid X_S = x_S\big], \tag{1}$$

so that missing features $X_{\bar{S}}$ are marginalized out using the conditional distribution $p(X_{\bar{S}} | X_S = x_s)$. Two special cases are $S = \varnothing$ and $S = D$, which respectively correspond to the mean prediction $f_\varnothing(x_\varnothing) = \mathbb{E}[f(X)]$ and the full model prediction $f_D(x) = f(x)$. This approach is common in recent work [1, 25] and is necessary for later connections with mutual information (Supplement A).

Using this convention for accommodating subsets of features, we can now measure how much $f$'s performance degrades when features are removed. Given a loss function $\ell$, the population risk for $f_S$ is defined as $\mathbb{E}\big[\ell\big(f_S(X_S), Y\big)\big]$ where the expectation is taken over the data distribution $p(X, Y)$. To define predictive power as a quantity that increases with model accuracy, we consider the *reduction in risk* over the mean prediction and define the function $v_f : \mathcal{P}(D) \mapsto \mathbb{R}$ as follows:

$$v_f(S) = \underbrace{\mathbb{E}\Big[\ell\big(f_\varnothing(X_\varnothing), Y\big)\Big]}_{\text{Mean prediction}} - \underbrace{\mathbb{E}\Big[\ell\big(f_S(X_S), Y\big)\Big]}_{\text{Using features } X_S}. \tag{2}$$

The domain is the power set $\mathcal{P}(D)$, the left term is the loss achieved with the mean prediction $\mathbb{E}[f(X)]$, and the right term is the loss achieved using the features $X_S$. The function $v_f(S)$ quantifies the amount of predictive power $f$ derives from the features $X_S$, and we generally expect that including more features in $S$ will make $v_f(S)$ larger. While $v_f$ provides a *model-based* notion of predictive power, we also introduce a notion of *universal predictive power*. For this, we define the function $v : \mathcal{P}(D) \mapsto \mathbb{R}$ as the reduction in risk from $X_S$ when using an optimal model:

$$v(S) = \underbrace{\min_{\hat{y}} \mathbb{E}\Big[\ell\big(\hat{y}, Y\big)\Big]}_{\text{Optimal constant } \hat{y}} - \underbrace{\min_{g} \mathbb{E}\Big[\ell\big(g(X_S), Y\big)\Big]}_{\text{Optimal model using } X_S}. \tag{3}$$

The left term is the loss from an optimal constant prediction $\hat{y}$ and the right term is the loss for an optimal model $g$ from the class of all functions (e.g., the Bayes classifier). Intuitively, $v$ represents the maximum predictive power that could hypothetically be derived from $X_S$. Since $f$ is typically trained using empirical risk minimization [40], the model-based predictive power $v_f$ provides an approximation to $v$ and the two coincide in certain cases where $f$ is optimal (Supplement B).

## 2.2 Additive Importance Measures

In certain very simple cases, features contribute predictive power in an additive manner. This means that we have $v(S \cup \{i\}) - v(S) = v(T \cup \{i\}) - v(T)$ for all subsets $S, T$ such that $i \notin S, T$. In these situations, we can define $X_i$'s importance as the predictive power it contributes, or $\phi_i = v(\{i\}) - v(\varnothing)$. However, a feature's contribution is generally not additive because it depends on which features $X_S$ are already present.

We therefore propose a class of *additive importance measures* that includes any method whose scores $\phi_1, \ldots, \phi_d$ can be understood as performance gains associated with each feature. This framework lets us unify numerous methods that either explicitly or implicitly define feature importance in terms of predictive power. The class of methods is defined as follows.

**Definition 1.** Additive importance measures *are methods that assign importance scores $\phi_i \in \mathbb{R}$ to features $i = 1, \ldots, d$ and for which there exists a constant $\phi_0 \in \mathbb{R}$ so that the additive function*

$$u(S) = \phi_0 + \sum_{i \in S} \phi_i$$

*provides a proxy for the predictive power of feature subsets, i.e., $u(S) \approx v(S)$.*

For methods in this class, $u(S)$ approximates $v(S)$ up to a constant value $\phi_0$ by summing the values $\phi_i$ for each included feature $i \in S$. Each $\phi_i$ can be viewed as the performance gain associated with $X_i$, which provides a measure of its importance for the prediction task. Although our definition focuses on methods that approximate the universal predictive power $v$, we also include methods that approximate model-based predictive power $v_f$ because these implicitly approximate $v$.

The function $v$ exhibits non-additive behavior for most prediction problems, so the proxy $u$ often cannot perfectly represent each feature's contribution to the predictive power. Although a crude approximation may provide users with some insight, closer approximations give a more accurate sense of each feature's importance. Next, we show that several existing methods manage this challenge by providing high quality approximations in specific regions of the domain $\mathcal{P}(D)$.

## 2.3 Existing Additive Importance Measures

Our framework of additive importance measures unifies numerous methods in the feature importance literature. These methods can be divided into three categories, representing the parts of the domain $\mathcal{P}(D)$ in which the additive proxy $u$ models $v$ or $v_f$ most accurately. Supplement G provides a table that summarizes the methods in each category.

The first category of methods characterize predictive power when no more than one feature is *excluded*, and they provide an additive function $u$ that accurately approximates $v$ or $v_f$ in the subdomain $\left( \{D\} \cup \{D \setminus \{i\} \mid i \in D\} \right) \subset \mathcal{P}(D)$. The canonical method for this is a feature ablation study where, in addition to a model $f$ trained on all features, separate models $f_1, f_2, \ldots, f_d$ are trained with individual features excluded [2, 15, 20]. Importance values $\phi_1, \ldots, \phi_d$ are then assigned based on the degradation in performance, according to the formula

$$\phi_i = \mathbb{E}\big[\ell\big(f_i(X_{D \setminus \{i\}}), Y\big)\big] - \mathbb{E}\big[\ell\big(f(X), Y\big)\big]. \tag{4}$$

A natural choice for $\phi_0$ in this case is $\phi_0 = \min_{\hat{y}} \mathbb{E}[\ell(\hat{y}, Y)] - \mathbb{E}[\ell(f(X), Y)] - \sum_{i \in D} \phi_i$, because then $u(D)$ and $u(D \setminus \{i\})$ approximate $v(D)$ and $v(D \setminus \{i\})$, respectively.

Several other methods provide similar notions of feature importance using a single model $f$. Permutation tests measure performance degradation when each column of the data is permuted [5]. Since permutation tests break feature dependencies, one variation suggests using a conditional permutation scheme [36]. Finally, in a method we refer to as "mean importance," performance degradation is measured after mean imputing individual features [32]. Although these methods take slightly different approaches, they all approximate either $v$ or $v_f$ when at most one feature is excluded.

The second category of methods describe $v$ when no more than one feature is *included*, providing an additive function $u$ that accurately describes $v$ in the subdomain $\left( \{\varnothing\} \cup \{\{i\} \mid i \in D\} \right) \subset \mathcal{P}(D)$.

Methods in this category model the bivariate association between $X_i$ and $Y$ to quantify $X_i$'s stand-alone predictive power. Bivariate association is commonly studied in fields such as computational biology (e.g., [23]) and is widely used to identify sensitive features [31]. As an example, the squared correlation $\text{Corr}(X_i, Y)^2$ is equivalent to the variance explained by a univariate linear model. More generally, one can measure the performance of univariate models trained to predict $Y$ given $X_i$ [13]. Given a model $g_i$ for each feature $i \in D$, the importance values are calculated with the formula

$$\phi_i = \min_{\hat{y}} \ \mathbb{E}\big[\ell\big(\hat{y}, Y\big)\big] - \mathbb{E}\big[\ell\big(g_i(X_i), Y\big)\big]. \tag{5}$$

A natural choice for the constant $\phi_0$ is $\phi_0 = 0$ in this case, because with these scores we see that $u(\varnothing) = v(\varnothing) = 0$ and that $u(\{i\})$ approximates $v(\{i\})$.

The two previous categories of methods provide imperfect notions of feature importance because they do not account for feature interactions. For example, two perfectly correlated features with significant predictive power would both be deemed unimportant by a feature ablation study, and two complementary features would have their importance underestimated by univariate models. The third category of methods addresses these issues by considering all feature subsets $S \subseteq D$.

Methods in the third category account for complex feature interactions by modeling $v$ across its entire domain $\mathcal{P}(D)$. These methods therefore supersede the two other categories, which either exclude or include individual features. Our method, SAGE, belongs to this category, and we show that SAGE assigns scores by modeling $v_f$ optimally via a weighted least squares objective (Section 3.2).

# 3 Shapley Additive Global Importance

We now introduce our method, Shapley additive global importance (SAGE), for quantifying a model's dependence on each feature. We present SAGE as an application of the game-theoretic Shapley value to $v_f$ and then examine its properties, including how it can be understood as an additive importance measure. Finally, we propose a practical sampling-based approximation algorithm.

## 3.1 Shapley Values for Credit Allocation

Recall that the function $v_f$ describes the amount of predictive power that a model $f$ derives from subsets of features $S \subseteq D$. We define feature importance via $v_f$ to quantify how critical each feature $X_i$ is for $f$ to make accurate predictions. It is natural to view $v_f$ as a cooperative game, representing the profit (predictive power) when each player (feature) participates (is available to the model). Research in game theory has extensively analyzed credit allocation for cooperative games, so we apply a game theoretic solution known as the Shapley value [33].

Shapley values are the unique credit allocation scheme that satisfies a set of fairness axioms. For any cooperative game $w : \mathcal{P}(D) \mapsto \mathbb{R}$ (such as $v$ or $v_f$) we may want the scores $\phi_i(w)$ assigned to each player to satisfy the following desirable properties:

1. (Efficiency) They sum to the total improvement over the empty set, $\sum_{i=1}^{d} \phi_i(w) = w(D) - w(\varnothing)$.
2. (Symmetry) If two players always make equal contributions, or $w(S \cup \{i\}) = w(S \cup \{j\})$ for all $S$, then $\phi_i(w) = \phi_j(w)$.
3. (Dummy) If a player makes zero contribution, or $w(S) = w(S \cup \{i\})$ for all $S$, then $\phi_i(w) = 0$.
4. (Monotonicity) If for two games $w$ and $w'$ a player always make greater contributions to $w$ than $w'$, or $w(S \cup \{i\}) - w(S) \geq w'(S \cup \{i\}) - w'(S)$ for all $S$, then $\phi_i(w) \geq \phi_i(w')$.
5. (Linearity) The game $w(S) = \sum_{k=1}^{n} c_k w_k(S)$, which is a linear combination of multiple games $(w_1, \ldots, w_n)$, has scores given by $\phi_i(w) = \sum_{k=1}^{n} c_k \phi_i(w_k)$.

The Shapley values $\phi_i(w)$ are the unique credit allocation scheme that satisfies properties 1-5 [33], and they are given by the expression:

$$\phi_i(w) = \frac{1}{d} \sum_{S \subseteq D \setminus \{i\}} \binom{d-1}{|S|}^{-1} \Big(w(S \cup \{i\}) - w(S)\Big). \tag{6}$$

The expression above shows that each Shapley value $\phi_i(w)$ is a weighted average of the incremental changes from adding $i$ to subsets $S \subseteq D \setminus \{i\}$. For SAGE, we propose assigning feature importance using the Shapley values of our model-based predictive power, or $\phi_i(v_f)$ for $i = 1, 2, \ldots, d$, which we refer to as *SAGE values*.

SAGE values satisfy many intuitive and desirable properties, arising both from the properties of Shapley values and from how $v_f$ is defined. These properties include:

1. Due to the efficiency property, SAGE values sum to $X$'s total predictive power. In other words, we have $\sum_{i=1}^{d} \phi_i(v_f) = v_f(D)$.

2. Due to the symmetry property, pairs of features $(X_i, X_j)$ with a deterministic relationship (e.g., perfect correlation) always have equal importance. To see this, remark that $f_{S \cup \{i\}}(x_{S \cup \{i\}}) = f_{S \cup \{j\}}(x_{S \cup \{j\}})$ for all $(S, x)$, so that $v_f(S \cup \{i\}) = v_f(S \cup \{j\})$.

3. Due to the dummy property, we have $\phi_i(v_f) = 0$ if $X_i$ is conditionally independent of $f(X)$ given all possible subsets of features $X_S$. However, features may receive non-zero importance even if they are not used by $f$ (which in some cases may be desirable).

4. Due to the monotonicity property, if we have two response variables $(Y, Y')$ with models $(f, f')$ and we have $v_f(S \cup \{i\}) - v_f(S) \geq v_{f'}(S \cup \{i\}) - v_{f'}(S)$ for all $S$, so that $X_i$ contributes more predictive power for $Y$ than for $Y'$, then the SAGE values satisfy $\phi_i(v_f) \geq \phi_i(v_{f'})$.

5. Due to the linearity property, SAGE values are the expectation of per-instance SHAP values applied to the model loss [24]. By this, we mean the Shapley values $\phi_i(v_{f,x,y})$ of the game

$$v_{f,x,y}(S) = \ell\big(f_\varnothing(x_\varnothing), y\big) - \ell\big(f_S(x_S), y\big).$$

In other words, the SAGE values are given by $\phi_i(v_f) = \mathbb{E}_{XY}[\phi_i(v_{f,X,Y})]$. These values were used in prior work, but they were not analyzed in depth and are costly to calculate via many local explanations [24].

6. Due to our definition of $v_f$, SAGE values are invariant to invertible mappings of the features. More precisely, if we apply an invertible function $h$ to a feature $X_i$ and define $Z_i = h(X_i)$, and we then use a model $f'$ that applies the inverse $h^{-1}$ to $Z_i$ before $f$, then the SAGE values of the new model under the new data distribution are unchanged. For example, SAGE values do not depend on whether a model uses gene counts or log gene counts.

Finally, SAGE values have an elegant interpretation when the loss function is cross entropy or mean squared error (MSE) and the model $f$ is optimal. With cross entropy loss, the optimal model (the Bayes classifier) predicts the conditional distribution $f^*(x) = p(Y|X = x)$ and the predictive power is $v_{f^*}(S) = \mathrm{I}(Y; X_S)$, where I denotes mutual information. The SAGE values are then given by:

$$\phi_i(v_{f^*}) = \frac{1}{d} \sum_{S \subseteq D \setminus \{i\}} \binom{d-1}{|S|}^{-1} \mathrm{I}(Y; X_i \mid X_S). \tag{7}$$

The expression above represents a weighted average of the conditional mutual information, i.e., the reduction in uncertainty for $Y$ when incorporating $X_i$ into different subsets $X_S$. An analogous result arises in the MSE case, and in both cases the SAGE values satisfy $\phi_i(v_{f^*}) \geq 0$ (Supplement C). Through this we see that although SAGE is a tool for model interpretation, it can also provide insight into intrinsic relationships in the data when applied with optimal models.

## 3.2 SAGE as an Additive Importance Measure

Though it not immediately obvious, SAGE is an additive importance measure (Section 2). Prior work has shown that Shapley values (Eq. 6) can be understood as the solution to a weighted least squares problem [6, 25]. From these findings, we see that SAGE provides an additive approximation to $v_f$ with $u(S) = \sum_{i \in S} \phi_i$, where $\phi_1, \ldots, \phi_d$ are optimal coefficients for the following problem:

$$\min_{\phi_1, \ldots, \phi_d} \sum_{S \subseteq D} \frac{d-1}{\binom{d}{|S|}|S|(d-|S|)} \left(\sum_{i \in S} \phi_i - v_f(S)\right)^2. \tag{8}$$

Understanding SAGE values in this way reveals that SAGE attempts to represent $v_f$ across its entire domain, modeling it optimally in a weighted least squares sense. Although the weights are perhaps not intuitive, this is the unique weighting scheme that leads to SAGE's desirable properties.

Using this interpretation of Shapley values, we observe that two more existing methods can be categorized as additive importance measures. The mean SHAP value of the loss [24] and Shapley Net Effects for linear models [22] are both similar to SAGE, but they involve more expensive calculations (explaining every individual prediction, or fitting an exponential number of linear models).

### 3.3 Practical SAGE Approximation

We now consider how to calculate SAGE values $\phi_i(v_f)$ efficiently. Obtaining these values is challenging because (i) there are an exponential number of subsets $S \subseteq D$ and (ii) evaluating a restricted model $f_S$ requires a Monte Carlo estimate with $X_{\bar{S}}$ sampled from $p(X_{\bar{S}}|X_S = x_S)$. Like previous methods that use Shapley values, we sidestep the inherent exponential complexity using an approximation algorithm [9, 18, 25, 37].

We address the first challenge by sampling random subsets of features $S \subseteq D$, and the second by sampling $X_{\bar{S}}$ from its marginal distribution. To efficiently generate subsets of features from the appropriate distribution, we enumerate over random permutations of the indices $D = \{1, \ldots, d\}$, similar to Štrumbelj and Kononenko [18]. Prior work has used sampling from the marginal distribution in a similar manner [25], but doing so alters some of SAGE's properties to align less with the information value of each feature and more with the model's mechanism. Supplement D describes the SAGE sampling algorithm (Algorithm 1) and the changes to its properties in more detail.

In Theorems 1 and 2, we make two claims regarding the estimates from our sampling algorithm (with proofs provided in Supplement E). The first result shows that the estimates converge to the correct values when run under appropriate conditions, and the second result shows that the estimates have variance that reduces at a linear rate.

**Theorem 1.** *The SAGE value estimates $\hat{\phi}_i(v_f)$ from Algorithm 1 converge to the correct values $\phi_i(v_f)$ when run with $n \to \infty$, $m \to \infty$, with an arbitrarily large dataset $\left\{(x^i, y^i)\right\}_{i=1}^{N}$, and with sampling from the correct conditional distribution $p(X_{\bar{S}}|X_S = x_S)$.*

**Theorem 2.** *The SAGE value estimates $\hat{\phi}_i(v_f)$ from Algorithm 1 have variance that reduces at the rate of $O(\frac{1}{n})$.*

In practice the algorithm must run for a finite number of iterations, so we propose an approach to monitor the estimates' uncertainty and detect convergence. Theorem 2 implies that for each feature $X_i$ there exists a constant $\sigma_i^2$ such that the estimate $\hat{\phi}_i(v_f)$ after $n$ iterations has variance given by $\text{Var}\big(\hat{\phi}_i(v_f)\big) \approx \sigma_i^2/n$. This quantity can be estimated while running the sampling algorithm and can then be used to provide confidence intervals on the estimated values. Finally, the algorithm may be considered converged when the largest standard deviation is a sufficiently low proportion $t$ (e.g., $t = 0.01$) of the range in the estimated values, or when the following criterion is satisfied:

$$\max_i \frac{\sigma_i}{\sqrt{n}} < t\Big(\max_i \hat{\phi}_i(v_f) - \min_i \hat{\phi}_i(v_f)\Big).$$

## 4 Related Work

Section 2 described prior work that was unified under the framework of additive importance measures, but there are also methods that do not fit into our framework. These are often model-specific heuristics that do not directly relate to the predictive power associated with each feature. For linear models, one heuristic is the magnitude of model coefficients [14]. For tree-based models, options include Gini importance and counting splits based on each feature [12]. And for neural networks, one can examine the magnitude of weights or aggregate local explanations [16], such as integrated gradients [38].

Shapley values have been studied extensively in game theory [33] and have been applied to machine learning for both local [7, 9, 25, 37] and global interpretability. For global interpretability, Shapley Net Effects proposed training linear models with every combination of features [22], which is similar

Table 1: Comparison of feature importance methods. *Agnostic:* method works with any model class. *Performance:* scores are related to the performance gains associated with each feature. *Interactions:* feature interactions are considered. *Missingness:* held out features are accounted for properly (e.g., by training a new model, or marginalizing them out). *Tractable:* method is computationally efficient. The symbol ✓ shows that a property is satisfied, × that it is not, and ∼ that it is to some extent.

| Method | AGNOSTIC | PERFORMANCE | INTERACTIONS | MISSINGNESS | TRACTABLE |
|---|---|---|---|---|---|
| Linear Model Coeff. Size | × | × | × | × | ✓ |
| Gini Importance | × | × | × | × | ✓ |
| Number of Splits | × | × | × | × | ✓ |
| Neural Network Weights | × | × | × | × | ✓ |
| Aggregated Local Saliency | × | × | × | × | ✓ |
| Mean Abs. SHAP | ✓ | × | ✓ | ∼ | × |
| Feature Ablation | ✓ | ✓ | × | ✓ | ∼ |
| Permutation Test | ✓ | ✓ | × | ∼ | ✓ |
| Conditional Permutation Test | ✓ | ✓ | × | ✓ | ✓ |
| Mean Importance | ✓ | ✓ | × | × | ✓ |
| Univariate Predictors | × | ✓ | × | ✓ | ✓ |
| Squared Correlation | × | ✓ | × | ✓ | ✓ |
| Shapley Net Effects | × | ✓ | ✓ | ✓ | × |
| Mean Loss SHAP | ✓ | ✓ | ✓ | ∼ | × |
| **SAGE** | ✓ | ✓ | ✓ | ∼ | ∼ |

to SAGE but often impractical due to the need for model retraining. Some work has considered applying Shapley values to function sensitivity, which is a subtly different problem than explaining the performance of machine learning models (Supplement F) [3, 27, 28, 35].

SHAP is a well-known method for local interpretability, but it has only been applied heuristically to global importance by calculating the mean absolute attribution value [25] and by using SHAP on the model loss rather than the model output [24]. SAGE draws a connection between global and local SHAP interpretability (Section 3.1), and our approach can converge hundreds or thousands of times faster than a naive calculation via SHAP because it avoids explaining individual instances in the dataset (Section 5).

Table 1 summarizes the related work by comparing a large number of methods. The table is separated into four groups, representing the categories of methods discussed in Section 2.3 and the additional methods discussed here. Only our approach, SAGE, satisfies each of the properties considered.

## 5   Experiments

We now evaluate SAGE by comparing it with several baseline methods. For simplicity we only consider model-agnostic baselines, including permutation tests, mean importance, feature ablation and univariate predictors (see Section 2.3). For datasets, we used MNIST [19], a bike sharing demand dataset [10], the German credit quality dataset [21], the Portuguese bank marketing dataset [26], and a breast cancer (BRCA) subtype classification dataset [4, 39]. We used XGBoost [8] for the bike data, CatBoost [29] for the bank and credit data, regularized logistic regression for the BRCA data, and a multi-layer perceptron (MLP) for MNIST. Supplement H provides more information about each dataset, including how we selected a subset of genes to avoid overfitting with the BRCA data.

Figure 1 shows examples of global explanations generated for MNIST and BRCA. For MNIST, most methods correctly assign high importance to the central region, but permutation tests and mean importance assign noisy scores with negative importance for many pixels in the central region. Feature ablation provides meaningless scores because removing individual features has a negligible impact on retrained models. SAGE and the univariate predictors approach provide the most plausible explanations, but their differences should be analyzed using quantitative metrics (see below).

The BRCA explanation (Figure 1 bottom) shows that genes previously known to be associated with BRCA receive the highest SAGE values. The most important gene (BCL11A) has a known association with a particularly aggressive form of BRCA [17], and the BRCA1 and BRCA2 genes are also highly ranked. Among the genes not associated with BRCA, the most highly ranked gene (SLC25A1) has a documented association with lung cancer [11].

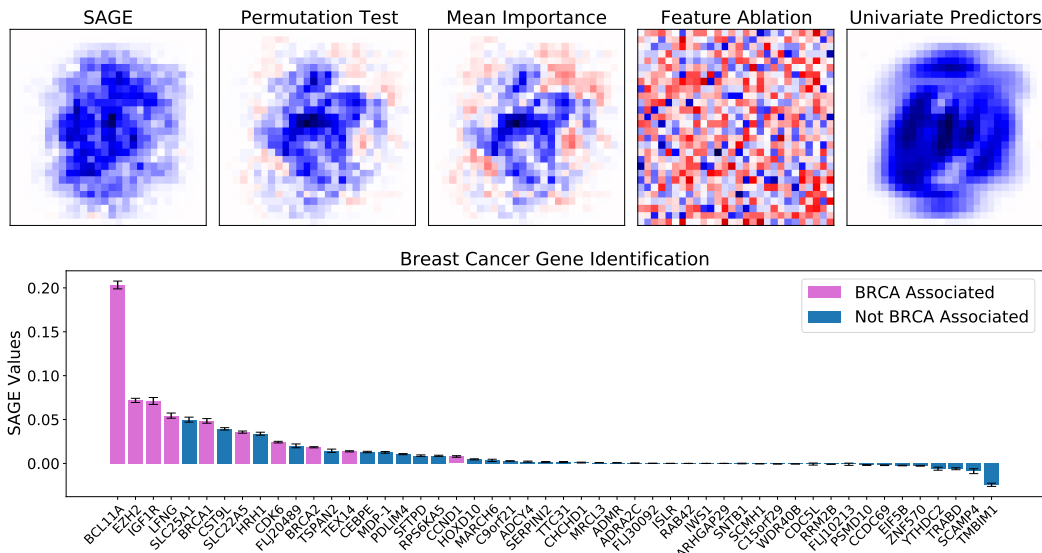

Figure 1: Global model explanations using SAGE. Top: comparison between SAGE and baseline methods for MNIST (blue pixels improve performance, red pixels hurt performance). Bottom: relevant gene identification using BRCA model (error bars indicate 95% confidence intervals).

We can also compare methods through quantitative metrics. As a first experiment, we seek to measure whether the sum of a method's importance scores is a reliable proxy for the predictive power of subsets of features. We generated several thousand feature subsets for each dataset, re-trained models for each subset, and then measured the correlation between the model loss and the total importance of the included features. Table 2 displays the results, which show that SAGE is either the best or nearly best for all datasets. Feature ablation fails for datasets with moderate numbers of features (BRCA and MNIST), but permutation tests and univariate predictors are both sometimes competitive.

Table 2: Correlation between total importance and performance of re-trained models.

|  | BANK MARKETING | BIKE DEMAND | CREDIT QUALITY | BRCA | MNIST |
|---|---|---|---|---|---|
| Permutation Test | 0.9918 | 0.9798 | **0.9571** | 0.8581 | 0.4855 |
| Mean Importance | – | 0.9798 | – | 0.8490 | 0.4824 |
| Feature Ablation | 0.9779 | 0.9651 | 0.2621 | -0.0371 | -0.4536 |
| Univariate | 0.9666 | 0.9587 | 0.9542 | **0.8718** | 0.4865 |
| SAGE | **0.9937** | **0.9815** | 0.9565 | 0.8611 | **0.4868** |

As another metric, we compared the predictive power of the highest and lowest ranked features according to each method. Figure 2 displays feature selection results on MNIST, where models are re-trained using the most (least) important features in the hopes of achieving high (low) accuracy. The results show that SAGE is most effective at identifying important features while univariate predictors are best for identifying unimportant features. Interestingly, most baselines fail to identify features with no predictive power (Figure 2 right), while SAGE performs well at both tasks.

Our experiments have focused thus far on accurate models, but SAGE can also be used to understand sub-optimal models. In a model monitoring context, SAGE can identify features that are corrupted or incorrectly encoded. Figure 3 shows an example where SAGE values are compared across several versions of the bank marketing dataset, including three test sets where the model accuracy decreased because a single feature is encoded inconsistently. When the month feature is incremented by one, the corresponding SAGE value is much lower than in the validation split; and when the call duration is no longer encoded in minutes, the corresponding SAGE values makes this change apparent.

From a computational perspective, our sampling algorithm (Section 3.3) is significantly faster than a naive calculation via local SHAP values. Our approach aims directly at a global explanation, whereas with SHAP one must generate and then average local explanations for hundreds or thousands of

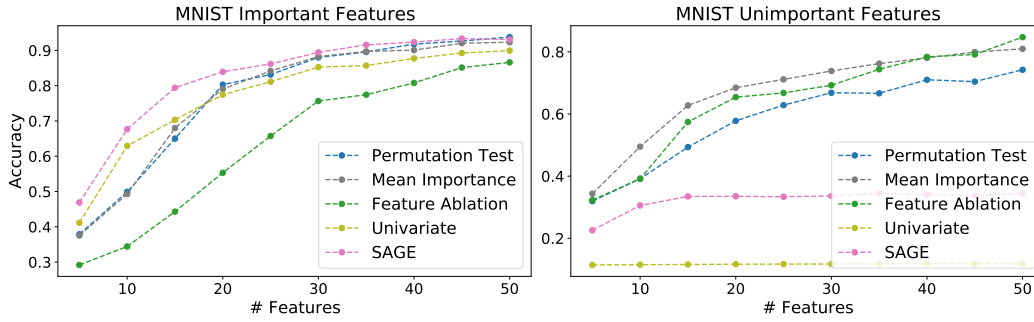

Figure 2: Feature selection with MNIST. Left: important features (higher accuracy is better). Right: unimportant features (lower accuracy is better).

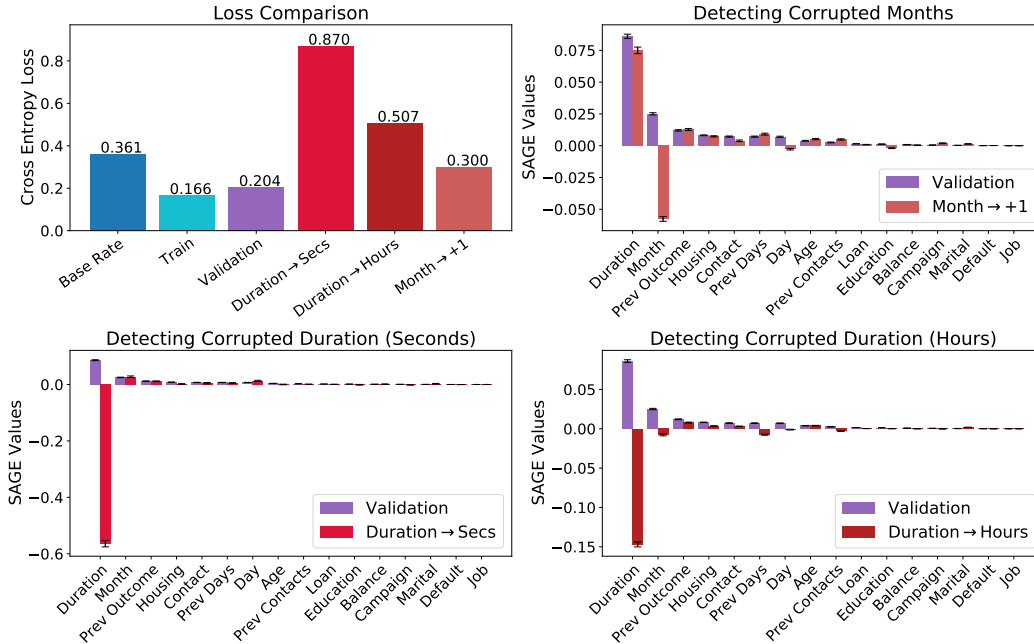

Figure 3: Identifying corrupted features with SAGE. Top left: performance comparison across dataset versions. Top right: SAGE comparison to identify corruption in month feature. Bottom left: SAGE comparison to identify corruption in duration feature (converted to seconds). Bottom right: SAGE comparison to identify corruption in duration feature (converted to hours).

examples. We found that the indirect approach using SHAP required 2-4 orders of magnitude more model evaluations to calculate the same values, which roughly corresponds to the dataset size (see Supplement H for details). This shows that SAGE is much faster for providing global explanations.

# 6  Conclusion

In this work we presented a unifying framework for global feature importance methods, as well as a new model-agnostic approach (SAGE) that accounts for complex feature interactions. Our perspective of quantifying predictive power shows that numerous existing methods make trade-offs regarding how to handle feature interactions; revealing each method's implicit assumptions allows users to reason explicitly about which tools to use. The method we introduced satisfies numerous desirable properties, and because of the sampling approach we propose, SAGE values can be calculated orders of magnitude faster than with a naive calculation via SHAP values. Future work will focus on estimating SAGE values even more efficiently and providing approximations that better model the conditional distribution of held out features.

## Broader Impact

This work contributes to a growing literature of methods that can provide researchers, engineers, and users with an understanding of how machine learning models work. By focusing on *global* explanations, our method produces succinct insights that the target audience may find more digestible than a large number of local explanations. Our work aims to contribute positive social impact, but as with any model explanation tool, there is a danger of deliberate misuse or adversarial use that could provide misleading information or be used to gain approval for bad practices.

## Acknowledgments and Disclosure of Funding

This work was funded by the National Science Foundation [CAREER DBI-1552309, and DBI-1759487]; the American Cancer Society [127332-RSG-15-097-01-TBG]; and the National Institutes of Health [R35 GM 128638, and R01 NIA AG 061132]. We would like to thank members of the Lee Lab and NeurIPS reviewers for feedback that greatly improved this work.

## Footnotes

[1] http://github.com/iancovert/sage/

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
