[Supplementary Material]

# Supplementary Materials for Understanding Global Feature Contributions With Additive Importance Measures

**Ian C. Covert**
University of Washington
Seattle, WA
icovert@uw.edu

**Scott Lundberg**
Microsoft Research
Redmond, WA
scott.lundberg@microsoft.com

**Su-In Lee**
University of Washington
Seattle, WA
suinlee@uw.edu

## A   Convention for Handling Missing Features

To examine how a model performs when deprived of certain information, we require a convention for evaluating $f$ with arbitrary subsets of features $S \subseteq D$. For this, we defined the restricted model $f_S$ as

$$f_S(x_S) = \mathbb{E}\big[f(X) \mid X_S = x_S\big].$$

Although we use an approximation in practice (Section 3.3), we have several reasons for defining SAGE using this convention. The reasons are: (i) the model $f_S$ is as close as possible to the full model $f$, (ii) the convention $f_S$ yields connections with intrinsic properties of the data distribution (such as mutual information), and (iii) alternative conventions often involve evaluating the model off the manifold of real data examples. We explain these points in detail below.

Consider how we can measure the deviation between a model $f$ on all features $X$ and a model $g$ on a subset of features $X_S$. We consider this separately for regression tasks and classification tasks. For a regression model $f$ that makes prediction in $\mathbb{R}^p$, a natural way to determine how much its prediction given $x$ differs from that of $g$ given $x_S$ is with the squared Euclidean distance $||f(x) - g(x_S)||^2$. In expectation, the squared deviation between $f$ and $g$ is equal to:

$$\mathbb{E}\Big[\big|\big|f(X) - g(X_S)\big|\big|^2\Big] = \underbrace{\mathbb{E}_X\Big[\big|\big|f(X) - \mathbb{E}[f(X) \mid X_S]\big|\big|^2\Big]}_{\text{Does not depend on } g} + \underbrace{\mathbb{E}_{X_S}\Big[\big|\big|\mathbb{E}[f(X) \mid X_S] - g(X_S)\big|\big|^2\Big]}_{\substack{\text{Mean squared deviation between} \\ g(X_S) \text{ and } \mathbb{E}[f(X) \mid X_S]}}$$

From this, it is clear that the model $g(x_S) = \mathbb{E}[f(X) \mid X_S = x_S]$ deviates least from $f$ on average.

Next, consider a classification model $f$ that outputs probabilities for a $p$-class categorical variable. To be explicit that the prediction is a vector of probabilities, we denote the model output as $f(y|x)$, where we have $f(i|x) \geq 0$ for $i = 1, 2, \ldots, p$ and $\sum_{i=1}^p f(i|x) = 1$. The Kullback-Leibler (KL) divergence $D_{\mathrm{KL}}\big(f(y|x) \mid\mid g(y|x_S)\big)$ is a natural way to measure the deviation of the predictions from $g$ and $f$. The mean deviation between $f$ and $g$ can then be expressed as:

$$\mathbb{E}_X\Big[D_{\mathrm{KL}}\big(f(y|X) \,\|\, g(y|X_S)\big)\Big] = \mathbb{E}_X\Big[\mathbb{E}_{y\sim f(y|X)}\big[-\log g(y|X_S)\big]\Big] - \mathbb{E}_X\Big[\mathrm{H}\big(f(y|X)\big)\Big]$$

$$= \mathbb{E}_{X_S}\Big[\mathbb{E}_{y\sim\mathbb{E}[f(y|X)\,|\,X_S]}\big[-\log g(y|X_S)\big]\Big]$$

$$\quad - \mathbb{E}_X\Big[\mathrm{H}\big(f(y|X)\big)\Big]$$

$$= \mathbb{E}_{X_S}\Big[\mathrm{H}\big(\mathbb{E}\big[f(y|X)\,|\,X_S\big]\big)\Big] - \mathbb{E}_X\Big[\mathrm{H}\big(f(y|X)\big)\Big]$$

$$\quad + \underbrace{\mathbb{E}_{X_S}\Big[D_{\mathrm{KL}}\big(\mathbb{E}\big[f(y|X)\,|\,X_S\big] \,\|\, g(y|X_S)\big)\Big]}_{\substack{\text{Mean KL divergence between}\\ g(y|X_S)\ \text{and}\ \mathbb{E}\big[f(y|X)\,|\,X_S\big]}}$$

With this way of rewriting the mean KL divergence, it becomes clear that the model $g(y|x_S) = \mathbb{E}[f(y|X)\,|\,X_S = x_S]$ is closest to $f$ in expectation.

These derivations show that in both cases, our convention for $f_S$ (marginalizing out features using their conditional distribution) yields the model that is closest to $f$ on average. When analyzing the performance degradation when $f$ is deprived of certain features, it is most conservative to use a restricted model $f_S$ that is as faithful to $f$ as possible, because doing otherwise may result in inflated estimates of the impact on model performance.

Next, handling missing features with $f_S$ yields connections with intrinsic properties of the data distribution, such as mutual information and conditional variance (Section C). That happens because when our convention is applied to an optimal model (e.g., the Bayes classifier), it preserves the model's optimality. For example, the Bayes classifier $f^*(x) = p(y|X = x)$ becomes the Bayes classifier $f_S^*(x_S) = p(y|X_S = x_S)$, and the conditional expectation $f^*(x) = \mathbb{E}[Y|X = x]$ becomes the conditional expectation $f_S^*(x_S) = \mathbb{E}[Y|X_S = x_S]$. Our definition of $f_S$ is the unique convention that has this property.

Finally, the convention $f_S$ only considers values of $x_{\bar{S}}$ such that $x = (x_S, x_{\bar{S}})$ has support under the full data distribution $p(X)$. That property is a benefit of handling missing features using their conditional distribution $p(X_{\bar{S}}|X_S = x_S)$. By contrast, other conventions may lead to implausible feature combinations with no support under the data distribution.

As an example, one alternative is to use the marginal distribution:

$$\mathbb{E}\big[f(x_S, X_{\bar{S}})\big].$$

This is what we do in practice (Section 3.3), but this breaks feature dependencies and may result in combinations of values $(x_S, x_{\bar{S}})$ that are off-manifold (e.g., if there are two perfectly correlated features and one is removed). We view this as an undesirable property and encourage work that removes the need for this approximation in practice.

Another option is to use the mean prediction when the missing features are drawn from the product of their marginal distributions, as in QII [1]. This convention is even more likely to result in off-manifold examples, because in addition to breaking dependencies between $X_S$ and $X_{\bar{S}}$, it also breaks dependencies within $X_{\bar{S}}$.

## B  Model-Based and Universal Predictive Power

In the main text, we introduce two set functions to represent different notions of predictive power. The function $v$ represents the *universal predictive power* and quantifies the amount of signal that can hypothetically be derived from a set of features $X_S$:

$$v(S) = \min_{\hat{y}}\ \mathbb{E}\Big[\ell\big(\hat{y}, Y\big)\Big] - \min_{g}\ \mathbb{E}\Big[\ell\big(g(X_S), Y\big)\Big].$$

In contrast, the function $v_f$ represents a *model-based* notion of predictive power and quantifies how much signal $f$ derives from a given set of features:

$$v_f(S) = \mathbb{E}\Big[\ell\big(f_\varnothing(X_\varnothing), Y\big)\Big] - \mathbb{E}\Big[\ell\big(f_S(X_S), Y\big)\Big].$$

The two quantities are different but related. To estimate $v(S)$, a natural approach would be to train a model using $X_S$, learn the optimal constant prediction $\hat{y}$, and then use the performance of those models as plug-in estimators for the two terms in $v(S)$. The model-based predictive power $v_f(S)$ can be viewed as an single-model approximation to this, where, instead of training a model from scratch on $X_S$, we obtain the model via an existing model $f$ trained using all features.

Under certain circumstances when the model $f^*$ is optimal, we can see that $v$ and $v_{f^*}$ coincide exactly for all $S \subseteq D$. Two simple cases where this holds are (i) for a regression task that uses the conditional expectation $f^*(x) = \mathbb{E}[Y|X = x]$ and mean squared error (MSE) loss, and (ii) for a classification task that uses the Bayes classifier $f^*(x) = p(y|X = x)$ and cross entropy loss. We show equality in the first case as follows:

$$\begin{aligned}
v_{f^*}(S) &= \mathbb{E}\Big[||Y - f^*_\varnothing(X_\varnothing)||^2\Big] - \mathbb{E}\Big[||Y - f^*_S(X_S)||^2\Big] \\
&= \mathbb{E}\Big[||Y - \mathbb{E}[Y]||^2\Big] - \mathbb{E}\Big[||Y - \mathbb{E}[Y|X_S = x_S]||^2\Big] \\
&= v(S)
\end{aligned}$$

Similarly, we show equality in the second case as follows:

$$\begin{aligned}
v_{f^*}(S) &= \mathbb{E}\big[-\log f^*_\varnothing(Y|X_\varnothing)\big] - \mathbb{E}\big[-\log f^*_S(Y|X_S)\big] \\
&= \mathbb{E}\big[-\log p(Y)\big] - \mathbb{E}\big[-\log p(Y|X_S = x_S)\big] \\
&= v(S)
\end{aligned}$$

Besides these two cases, equality between $v$ and $v_{f^*}$ holds for optimal models $f^*$ when using a specific class of loss function— loss functions $\ell$ that satisfy the property that an optimal model $f^*$ for $X$ yields an optimal model $f^*_S$ for $X_S$. For example, this property holds for all strictly proper scoring rules, because optimal models under these loss functions are probabilistic forecasts [2]

## C  SAGE Properties with Optimal Models

### C.1  Properties with Bayes Classifier

Here, we derive the properties of SAGE when it is applied to the Bayes classifier with cross entropy loss. We derive the claim from scratch, beginning with a proof that the Bayes classifier is optimal. To be explicit that the prediction is a vector of probabilities, we denote the model output as $f(y|x)$, where we have $f(i|x) \geq 0$ for $i = 1, 2, \ldots, p$ and $\sum_{i=1}^p f(i|x) = 1$. We also use H to denote entropy and I to denote mutual information.

For a classification model trained with cross entropy loss, we decompose the population risk as follows to reveal the optimal classifier:

$$\begin{aligned}
\mathbb{E}\big[\ell\big(f(y|X), Y\big)\big] &= \mathbb{E}\big[-\log f(Y|X)\big] \\
&= \mathbb{E}_X\Big[\mathbb{E}_{Y|X}\big[-\log f(Y|X)\big]\Big] \\
&= \mathbb{E}_X\big[D_{\mathrm{KL}}\big(p(y|X) \,||\, f(y|X)\big)\big] + \mathrm{H}(Y|X)
\end{aligned}$$

The entropy term is constant, so the optimal prediction model is the Bayes classifier $f^*(x) = p(y|X = x)$. We now consider the application of SAGE to the model $f^*$. The restricted models $f^*_S$ are the following:

$$f_S^*(y|x_S) = \mathbb{E}\big[f^*(y|X) \mid X_S = x_S\big]$$
$$= \mathbb{E}\big[p(y|X) \mid X_S = x_S\big]$$
$$= p(y|X_S = x_S)$$

The risk incurred by the restricted model $f_S^*$ is then:

$$\mathbb{E}\big[\ell\big(f_S^*(y|X_S), Y\big)\big] = \mathbb{E}\big[-\log f_S^*(Y|X_S)\big]$$
$$= \mathbb{E}\big[-\log p(Y|X_S)\big]$$
$$= \mathrm{H}(Y|X_S)$$

We can now see that the cooperative game $v_{f^*}$ is:

$$v_{f^*}(S) = \mathbb{E}\big[\ell\big(f_{\varnothing}^*(X_{\varnothing}), Y\big)\big] - \mathbb{E}\big[\ell\big(f_S^*(X_S), Y\big)\big]$$
$$= \mathrm{H}(Y) - \mathrm{H}(Y \mid X_S)$$
$$= \mathrm{I}(Y; X_S)$$

In the expression for Shapley values, the weighted summation has terms of the following form:

$$v(S \cup \{i\}) - v(S) = \mathrm{I}(Y; X_{S \cup \{i\}}) - \mathrm{I}(Y; X_S)$$
$$= \mathrm{H}(Y \mid X_S) - \mathrm{H}(Y \mid X_{S \cup \{i\}})$$
$$= \mathrm{I}(Y; X_i \mid X_S)$$

This completes the derivation for the result in the main text, because we see that the Shapley values are equal to

$$\phi_i(v_{f^*}) = \frac{1}{d} \sum_{S \subseteq D \setminus \{i\}} \binom{d-1}{|S|}^{-1} \mathrm{I}(Y; X_i \mid X_S).$$

### C.2 Properties with Conditional Expectation

We now show a similar result for optimal regression models when using mean squared error (MSE) loss. We assume that the predictions are scalars, although a similar result holds for vector-valued predictions. We first decompose the population risk to determine the optimal model:

$$\mathbb{E}\big[\ell\big(f(X), Y\big)\big] = \mathbb{E}\big[\big(f(X) - Y\big)^2\big]$$
$$= \mathbb{E}\big[\big(f(X) - \mathbb{E}[Y|X]\big)^2\big] + \mathbb{E}\big[\big(\mathbb{E}[Y|X] - Y\big)^2\big]$$

Only the first term depends on $f$, so it is clear that the conditional expectation function $f^*(x) = \mathbb{E}[Y|X = x]$ is optimal. We now consider the application of SAGE to the model $f^*$. The restricted models $f_S^*$ are the following:

$$f_S^*(x_S) = \mathbb{E}\big[f^*(X) \mid X_S = x_S\big]$$
$$= \mathbb{E}\big[\mathbb{E}[Y|X] \mid X_S = x_S\big]$$
$$= \mathbb{E}[Y|X_S = x_S]$$

The last line follows from the law of iterated expectations. The risk incurred by the restricted model $f_S^*$ is then:

$$\mathbb{E}\big[\ell\big(f_S^*(X_S), Y\big)\big] = \mathbb{E}\big[\big(\mathbb{E}[Y \mid X_S] - Y\big)^2\big]$$
$$= \mathbb{E}[\mathrm{Var}(Y|X_S)]$$

We now see that the cooperative game $v_{f^*}$ is:

$$v_{f^*}(S) = \mathrm{Var}(Y) - \mathbb{E}\big[\mathrm{Var}(Y|X_S)\big]$$
$$= \mathrm{Var}\big(\mathbb{E}[Y|X]\big)$$

The last line follows from the law of total variance. The difference terms in the Shapley summation are the following:

$$v_{f^*}(S \cup \{i\}) - v_{f^*}(S) = \mathbb{E}\big[\mathrm{Var}(Y|X_S)\big] - \mathbb{E}\big[\mathrm{Var}(Y|X_{S\cup\{i\}})\big]$$
$$= \mathbb{E}_{X_S}\Big[\mathrm{Var}\big(\mathbb{E}[Y|X_S, X_i] \mid X_S\big)\Big]$$

The last line also follows from the law of total variance. Intuitively, these terms quantify the average amount of variation left in the random variable $\mathbb{E}[Y \mid X_S, X_i]$ when $X_i$ is unknown but distributed according to $p(X - i|X_S)$. If the amount of variation is high, then $i$ contains significant incremental information about $Y$. The above expression is analogous to $\mathrm{I}(Y; X_i \mid X_S)$ from the classification case.

Finally, we see that the SAGE values are equal to

$$\phi_i(v_{f^*}) = \frac{1}{d} \sum_{S \subseteq D \setminus \{i\}} \binom{d-1}{|S|}^{-1} \mathbb{E}\Big[\mathrm{Var}\big(\mathbb{E}[Y|X_S, X_i]\big|X_S\big)\Big].$$

## D  SAGE Sampling

Here, we describe our sampling algorithm in greater detail and discuss the consequences of using the marginal rather than conditional distribution.

### D.1  SAGE Sampling Algorithm

Our sampling approach is displayed below in Algorithm 1. In Algorithm 1, each feature's score $\phi_i(v_f)$ is estimated by attempting to average many samples of the form $\ell\big(f_S(x_S), y\big) - \ell\big(f_{S\cup\{i\}}(x_{S\cup\{i\}}), y\big)$, or the incremental improvement in the loss when incorporating $X_i$ for a particular input-label pair $(x, y)$. In each sample, we draw $(x, y)$ from the empirical data distribution, we determine $S$ based on a random permutation $\pi$ of feature indices $D$, and we estimate $f_S(x_S)$ via Monte Carlo approximation with a distribution $q(X_{\bar{S}}|X_S = x_S)$ substituted for $p(X_{\bar{S}}|X_S = x_S)$.

One practical option for the distribution $q$, which we use in our experiments, is to sample from the marginal distribution $q(X_{\bar{S}}|X_S = x_S) = p(X_{\bar{S}})$, which corresponds to an assumption of feature independence. Another option is to mean impute the missing features, which corresponds to a further assumption of model linearity [4].

### D.2  Properties with Marginal Sampling

SAGE's properties change when Algorithm 1 is run with sampling from the marginal distribution $p(X_{\bar{S}})$ rather than the conditional distribution $q(X_{\bar{S}}|X_S = x_S)$. Sampling from the marginal instead of the conditional changes the underlying cooperative game and means that we no longer estimate the Shapley values $\phi_i(v_f)$, but rather the Shapley values of a different game.

Sampling from the marginal distribution means that in the inner loop of Algorithm 1, the Monte Carlo approximation estimates $\mathbb{E}[f(x_S, X_{\bar{S}})]$. We adopt the notation $\tilde{f}_S$ to denote an alternative restricted model (instead of $f_S$):

---
**Algorithm 1** Sampling-based approximation for SAGE values
---
    **Input:** data $\{x^i, y^i\}_{i=1}^N$, model $f$, loss function $\ell$, outer samples $n$, inner samples $m$

    Initialize $\hat{\phi}_1 = 0, \hat{\phi}_2 = 0, \ldots, \hat{\phi}_d = 0$

    marginalPred $= \frac{1}{N} \sum_{i=1}^N f(x_i)$

    **for** $i = 1$ **to** $n$ **do**

        Sample $(x, y)$ from $\{x^i, y^i\}_{i=1}^N$

        Sample $\pi$, a permutation of $D$

        $S = \varnothing$

        lossPrev $= \ell(\text{marginalPred}, y)$

        **for** $j = 1$ **to** $d$ **do**

            $S = S \cup \{\pi[j]\}$

            $y = 0$

            **for** $k = 1$ **to** $m$ **do**

                Sample $x_{\bar{S}}^k \sim q(x_{\bar{S}} | X_S = x_S)$

                $y = y + f(x_S, x_{\bar{S}}^k)$

            **end for**

            $\bar{y} = \frac{y}{m}$

            loss $= \ell(\bar{y}, y)$

            $\Delta = \text{lossPrev} - \text{loss}$

            $\hat{\phi}_{\pi[j]} = \hat{\phi}_{\pi[j]} + \Delta$

            lossPrev $= \text{loss}$

        **end for**

    **end for**

    **return** $\frac{\hat{\phi}_1}{n}, \frac{\hat{\phi}_2}{n}, \ldots, \frac{\hat{\phi}_d}{n}$
---

$$\tilde{f}_S(x_S) = \mathbb{E}[f(x_S, X_{\bar{S}})].$$

We then adopt the notation $\tilde{v}_f$ to denote the new cooperative game using $\tilde{f}_S$, which is

$$\tilde{v}_f(S) = \mathbb{E}\Big[\ell\big(\tilde{f}_\varnothing(X_\varnothing), Y\big)\Big] - \mathbb{E}\Big[\ell\big(\tilde{f}_S(X_S), Y\big)\Big].$$

Sampling from the marginal distribution means that we estimate the Shapley values $\phi_i(\tilde{v}_f)$ instead of $\phi_i(v_f)$. Some of the properties of the importance scores $\phi_i(\tilde{v}_f)$ differ from those described in Section 3.1:

1. Due to the efficiency property, the scores satisfy $\sum_{i=1}^d \phi_i(\tilde{v}_f) = \tilde{v}_f(D) - \tilde{v}_f(\varnothing)$. Because we have $\tilde{f}_\varnothing(x_\varnothing) = f_\varnothing(x_\varnothing)$ and $\tilde{f}_D(x) = f_D(x)$ for all $x$ (i.e., the models are the same given either all features or no features), we also have $\sum_{i=1}^d \phi_i(\tilde{v}_f) = \sum_{i=1}^d \phi_i(v_f) = v_f(D)$.

2. Due to the symmetry property, we have $\phi_i(\tilde{v}_f) = \phi_j(\tilde{v}_f)$ when $\tilde{v}_f(S \cup \{i\}) = \tilde{v}_f(S \cup \{j\})$ for all $S$. That holds if we have $\tilde{f}_{S \cup \{i\}}(x_{S \cup \{i\}}) = \tilde{f}_{S \cup \{j\}}(x_{S \cup \{j\}})$ for all $(S, x)$. Given our definition of $\tilde{f}_S$, there is no simple sufficient condition for when this holds for $(X_i, X_j)$. Unlike in the original formulation, perfectly correlated features may not receive equal importance.

3. Due to the dummy property, we have $\phi_i(\tilde{v}_f) = 0$ if $\tilde{f}_S(x_S) = \tilde{f}_{S \cup \{i\}}(x_{S \cup \{i\}})$ for all $(S, x)$. A sufficient condition for this to hold is that the model $f$ is invariant to $X_i$. That means that the value $\phi_i(\tilde{v}_f)$ for a sensitive attribute $X_i$ (e.g., race) may be zero even if the model depends on correlated features (e.g., zip code).

4. Due to the monotonicity property, if we have two response variables $Y, Y'$ with models $f, f'$, and we have $\tilde{v}_f(S \cup \{i\}) - \tilde{v}_f(S) \geq \tilde{v}_{f'}(S \cup \{i\}) - \tilde{v}_{f'}(S)$ for all $S$, then we have $\phi_i(\tilde{v}_f) \geq \phi_i(\tilde{v}_{f'})$. This means that if $X_i$ contributes more predictive power to $Y$ than to $Y'$, then it receives more importance for $Y$.

5. Due to the linearity property, the values $\phi_i(\tilde{v}_f)$ are the expectation of per-instance loss SHAP values computed using the marginal distribution. If we define the game $\tilde{v}_{f,x,y}$ as

$$\tilde{v}_{f,x,y}(S) = \ell\big(\tilde{f}_{\varnothing}(x_{\varnothing}), y\big) - \ell\big(\tilde{f}_{S}(x_{S}), y\big),$$

then we have $\phi_i(\tilde{v}_f) = \mathbb{E}_{XY}\big[\phi_i(\tilde{v}_{f,X,Y})\big]$. These are loss SHAP values computed with the same feature independence assumption [4].

6. As in the original formulation, the values $\phi_i(\tilde{v}_f)$ are invariant to invertible transformations of the features.

One elegant aspect of the original formulation of SAGE is a connection with intrinsic properties of the data distribution when applied with optimal models (e.g., the Bayes classifier). We lose these connections when using the marginal distribution because the restricted models $\tilde{f}_S^*$ based on optimal models $f^*$ are no longer optimal for $X_S$. This formulation is therefore less aligned with the information value of each feature.

However, some recent work has advocated for advantages of sampling from the marginal distribution in SHAP [3]. The most appealing property is that features that are not used by the model always receive zero attribution, a property that we showed also holds for SAGE.

## E  Proofs

The two results from Section 3.3 of the main text are restated and proved below.

**Theorem 1.** *The SAGE value estimates $\hat{\phi}_i(v_f)$ from Algorithm 1 converge to the correct values $\phi_i(v_f)$ when run with $n \to \infty$, $m \to \infty$, with an arbitrarily large dataset $\{(x^i, y^i)\}_{i=1}^{N}$, and with sampling from the correct conditional distribution $p(X_{\bar{S}}|X_S = x_S)$.*

*Proof.* At a high level, the algorithm has an outer loop that contributes one sample to each of the SAGE value estimates $\hat{\phi}_i(v_f)$. Each estimate can be interpreted as a sample mean that converges to its expectation as $n$ becomes large. Our proof proceeds by considering the value of the expectation under the assumptions that $m \to \infty$ and $q(X_{\bar{S}}|X_S = x_S) = p(X_{\bar{S}}|X_S = x_S)$.

Each estimate $\hat{\phi}_i(v_f)$ is the average of many samples of the random variable $\Delta_{X,Y,S}^{i,m}$ which we define here as

$$\Delta_{x,y,S}^{i,m} = \ell\Big(\frac{1}{m}\sum_{k=1}^{m} f(x_S, x_{\bar{S}}^k), y\Big) - \ell\Big(\frac{1}{m}\sum_{l=1}^{m} f(x_{S\cup\{i\}}, x_{\bar{S}\setminus\{i\}}^l), y\Big). \tag{1}$$

Specifically, we have

$$\hat{\phi}_i(v_f) = \frac{1}{n}\sum_{j=1}^{n} \Delta_{x_j, y_j, S_j}^{i,m} \tag{2}$$

where $i, m$ are fixed and $x_j, y_j$ and $S_j$ are determined by each iteration of Algorithm 1. Even for fixed $i, m, x, y, S$, note that $\Delta_{x,y,S}^{i,m}$ is a random variable because each $x_{\bar{S}}^k$ and $x_{\bar{S}\setminus\{i\}}^l$ are independent samples from the distributions $q(X_{\bar{S}}|X_S = x_S)$ and $q(X_{\bar{S}\setminus\{i\}}|X_{S\cup\{i\}} = x_{S\cup\{i\}})$ respectively. We begin by analyzing the random variable $\Delta_{x,y,S}^{i,m}$ and what it converges to as $m \to \infty$.

Consider the first term in Eq. 1. The mean prediction $\frac{1}{m}\sum_{k=1}^{m} f(x_S, x_{\bar{S}}^k)$ provides a Monte Carlo approximation of

$$\mathbb{E}_{q(X_{\bar{S}}|X_S = x_S)}[f(x_S, X_{\bar{S}})].$$

We assume that samples are from the true conditional distribution $p(X_{\bar{S}}|X_S = x_S)$, so the average prediction $\frac{1}{m}\sum_{k=1}^{m} f(x_S, x_{\bar{S}}^k)$ is in fact an approximation of $f_S(x_S)$. When we let $m \to \infty$ the law of large numbers says that

$$\frac{1}{m}\sum_{k=1}^{m} f(x_S, x_{\bar{S}}^k) \xrightarrow{p} f_S(x_S) \tag{3}$$

where $\xrightarrow{p}$ denotes convergence in probability. By an identical argument for the second term in Eq. 1, because of sampling from $p(X_{\bar{S}\setminus\{i\}}|X_{S\cup\{i\}} = x_{S\cup\{i\}})$, we see that

$$\frac{1}{m}\sum_{l=1}^{m} f(x_{S\cup\{i\}}, x_{\bar{S}\setminus\{i\}}^l) \xrightarrow{p} f_{S\cup\{i\}}(x_{S\cup\{i\}}) \tag{4}$$

as $m \to \infty$. This lets us conclude that for fixed $x, y, S$, $\Delta_{x,y,S}^{i,m}$ converges as $m \to \infty$ as follows:

$$\Delta_{x,y,S}^{i,m} \xrightarrow{p} \ell\big(f_S(x_S), y\big) - \ell\big(f_{S\cup\{i\}}(x_{S\cup\{i\}}), y\big). \tag{5}$$

With this result, we define $\Delta_{x,y,S}^{i} \equiv \lim_{m\to\infty} \Delta_{x,y,S}^{i,m}$. We now consider the fact that the final estimates $\hat{\phi}_i(v_f)$ are the average of many samples $\Delta_{X,Y,S}^{i,m}$, or many samples $\Delta_{X,Y,S}^{i}$ in the limit $m \to \infty$. We will determine the expected value of $\hat{\phi}_i(v_f)$ and argue that it converges to this value as $n \to \infty$.

First, consider the distribution from which $S$ is implicitly drawn. In Algorithm 1, $S$ is determined by a permutation $\pi$ of the feature indices $D = \{1, \ldots, d\}$ and it contains indices that are already included when we arrive at feature $i$. The number of indices $|S|$ preceding $i$ is uniformly distributed between 0 and $d - 1$, and the preceding indices are chosen uniformly at random among the $\binom{d-1}{|S|}$ possible combinations. We can therefore write a probability mass function $p(S)$ for subsets $S$ that may be included by the time when $i$ is added:

$$p(S) = \frac{1}{d}\binom{d-1}{|S|}^{-1}. \tag{6}$$

When we take the expectation of $\Delta_{x,y,S}^{i}$ over $S$, we have

$$\mathbb{E}_{p(S)}\Big[\Delta_{x,y,S}^{i}\Big] = \mathbb{E}_{p(S)}\Big[\ell\big(f_S(x_S), y\big) - \ell\big(f_{S\cup\{i\}}(x_{S\cup\{i\}}), y\big)\Big]$$
$$= \sum_{T \subseteq D\setminus\{i\}} \frac{1}{d}\binom{d-1}{|T|}^{-1}\Big(\ell\big(f_T(x_T), y\big) - \ell\big(f_{T\cup\{i\}}(x_{T\cup\{i\}}), y\big)\Big). \tag{7}$$

The expression above already resembles the Shapley value because of the weighted summation over subsets. We can now incorporate an expectation over $(x, y)$ pairs drawn from the data distribution $p(X, Y)$ and see that the Shapley value $\phi_i(v_f)$ arises naturally:

$$\mathbb{E}_{XY}\mathbb{E}_{p(S)}\Big[\Delta_{X,Y,S}^{i}\Big] = \mathbb{E}_{XY}\mathbb{E}_{p(S)}\Big[\ell\big(f_S(X_S), Y\big) - \ell\big(f_{S\cup\{i\}}(X_{S\cup\{i\}}), Y\big)\Big]$$
$$= \mathbb{E}_{p(S)}\big[v_f(S \cup \{i\}) - v_f(S)\big]$$
$$= \phi_i(v_f). \tag{8}$$

Finally, we invoke the law of large numbers again to conclude that in the limit of an arbitrarily large dataset $\{(x^i, y^i)\}_{i=1}^{N}$ drawn from $p(X, Y)$, we have the convergence result

$$\hat{\phi}_i(v_f) \xrightarrow{p} \phi_i(v_f) \tag{9}$$

as $n \to \infty, m \to \infty$.

In summary, our proof is the following:

$$\hat{\phi}_i(v_f) = \frac{1}{n} \sum_{j=1}^{n} \Delta_{x_j, y_j, S_j}^{i,m}$$

$$\Delta_{x,y,S}^{i} \equiv \lim_{m \to \infty} \Delta_{x,y,S}^{i,m}$$

$$= \ell\big(f_S(x_S), y\big) - \ell\big(f_{S \cup \{i\}}(x_{S \cup \{i\}}), y\big)$$

$$\lim_{m \to \infty} \hat{\phi}_i(v_f) = \frac{1}{n} \sum_{j=1}^{n} \Delta_{x_j, y_j, S_j}^{i}$$

$$\lim_{n \to \infty} \lim_{m \to \infty} \hat{\phi}_i(v_f) = \mathbb{E}_{XY} \mathbb{E}_{p(S)} \big[ \Delta_{X,Y,S}^{i} \big]$$

$$= \phi_i(v_f)$$

$\square$

We now prove the second result.

**Theorem 2.** *The SAGE value estimates $\hat{\phi}_i(v_f)$ from Algorithm 1 have variance that reduces at the rate of $O(\frac{1}{n})$.*

*Proof.* At a high level, the algorithm has an outer loop that contributes one sample $\Delta_{X,Y,S}^{i,m}$ (see Eq. 1) to each of the SAGE estimates $\hat{\phi}_i(v_f)$, where randomness arises from the sampling of $x, y$ and $S$ and the inner loop samples (see Eq. 2). Regardless of how $q(X_{\bar{S}}|X_S = x_S)$ is chosen and the number of inner loop samples $m$, the central limit theorem says that as $n$ becomes large, the sample mean $\hat{\phi}_i(v_f)$ converges in distribution to a Gaussian with mean $\mathbb{E}[\Delta_{X,Y,S}^{i,m}]$ and variance equal to

$$\frac{\text{Var}(\Delta_{X,Y,S}^{i,m})}{n}. \tag{10}$$

Although we do not have access to the numerator $\text{Var}(\Delta_{X,Y,S}^{i,m})$, we can conclude that the variance of the estimates behaves as $O(\frac{1}{n})$.

$\square$

# F    Function Sensitivity

Several recent papers considered a related question of a function's sensitivity to its various inputs [5, 6, 7]. We briefly describe the problem to illustrate how it differs from our work.

For a scalar real-valued function $f$ defined on multiple features $x = (x_1, x_2, \ldots, x_d)$, this body of work seeks to assign a sensitivity value to each feature. This is done with a variance-based measure of the dependence of $f$ on each feature, through an analysis of the cooperative game

$$w_f(S) = \text{Var}\big(f(X)\big) - \mathbb{E}\big[\text{Var}(f(X) \mid X_S)\big]$$

$$= \text{Var}\big(\mathbb{E}[f(X) \mid X_S]\big). \tag{11}$$

The Shapley values $\phi_i(w_f)$ serve as sensitivity measures for each feature $i = 1, 2, \ldots, d$. In prior work, Owen connected this measure of feature importance to the two Sobol' indices [5], Owen and Prieur considered special cases with closed form solutions [6], and Song et al. provided a sampling-based approximation algorithm [7].

Our work considers the problem of assessing feature importance for black-box machine learning models. In contrast with the function sensitivity work, we allow for a response variable $Y$ that is

jointly distributed with $X$, which is not necessarily in $\mathbb{R}$, and we consider how predictive each feature is of $Y$. The work on function sensitivity is equivalent to an application of SAGE to the special case where $Y \equiv f(X)$, where the model output is real-valued $f(X) \in \mathbb{R}$, and where the loss $\ell$ is MSE loss.

For cases with a response variable $Y \in \mathbb{R}$, a natural question is whether there is a relationship between $\phi_i(v_f)$ and $\phi_i(w_f)$. The only case when these values coincide is when the loss is MSE and the model $f$ is the conditional expectation $f^*(x) = \mathbb{E}[Y|X = x]$. Equality of the Shapley values follows from equality of the cooperative games:

$$
\begin{aligned}
w_{f^*}(S) &= \mathrm{Var}\big(\mathbb{E}[f^*(X) \mid X_S]\big) \\
&= \mathrm{Var}\big(\mathbb{E}[\mathbb{E}[Y \mid X] \mid X_S]\big) \\
&= \mathrm{Var}\big(\mathbb{E}[Y \mid X_S]\big) \\
&= \mathrm{Var}(Y) - \mathbb{E}[\mathrm{Var}(Y \mid X_S)] \\
&= \mathbb{E}\big[\big(Y - f_\varnothing^*(X_\varnothing)\big)\big] - \mathbb{E}\big[\big(Y - f_S^*(X_S)\big)^2\big] \\
&= v_{f^*}(S)
\end{aligned}
$$

Outside of this special case, the sensitivity values $\phi_i(w_f)$ differ from the SAGE values $\phi_i(v_f)$.

## G  Summary of Additive Importance Measures

Table 1 summarizes the additive importance measures described in Section 2.3. For each method, we indicate which part of the subdomain of $\mathcal{P}(D)$ it prioritizes and show whether it approximates $v$ (universal predictive power) or $v_f$ (model-based predictive power).

Permutation tests, conditional permutation tests and mean importance assign scores in a similar manner. Each of these methods make different assumptions to approximate the difference

$$
v_f(D \setminus \{i\}) - v_f(D) = \mathbb{E}\Big[\ell\big(f_{D\setminus\{i\}}(X_{D\setminus\{i\}}), Y\big)\Big] - \mathbb{E}\Big[\ell\big(f(X), Y\big)\Big],
$$

where the restricted model is given by:

$$
f_{D\setminus\{i\}}(x_{D\setminus\{i\}}) = \mathbb{E}_{X_i | X_{D\setminus\{i\}}}\big[f(x_{D\setminus\{i\}}, X_i)\big].
$$

Conditional permutation tests make the closest approximation, but they have the expectation over $X_i$ outside the loss function instead of inside it. Permutation tests make an assumption of feature independence by sampling $X_i$ from its marginal distribution $p(X_i)$ instead of from its conditional distribution $p(X_i | X_{D\setminus\{i\}} = x_{D\setminus\{i\}})$. And finally, mean importance makes a further assumption of model linearity by simply using the marginal mean $\mathbb{E}[X_i]$.

## H  Experiment Details

This section provides more details about our experiments, as well as several additional results.

### H.1  Datasets and Hyperparameters

Table 2 provides information about the datasets, models and validation/test splits used in our experiments. All hyperparameters were chosen using the validation data (number of rounds for decision trees, $\ell_1$ regularization for BRCA, early stopping for the MLP) and the test data were reserved for measuring model performance and calculating feature importance.

The BRCA data required a small amount of pre-processing. We manually selected a small set of genes with known BRCA associations and then selected the remaining genes uniformly at random to have 50 total genes. A small number of missing expression values were imputed with their mean.

Table 1: Summary of additive importance measures. *Approximates*: whether the method approximates universal or model-based predictive power. *Importance values*: the values assigned as each method is run until convergence, with $\phi_0$ indicating the value for which $u(S)$ closely approximates $v$ or $v_f$. For Shapley Net Effects and SAGE $\phi_i(\cdot)$ denotes the Shapley value.

| SUBDOMAIN | APPROXIMATES | METHOD | IMPORTANCE VALUES |
|---|---|---|---|
| $\{D\} \cup \{\{D \setminus \{i\}\} \mid i \in D\}$ | $v$ | Feature Ablation | $\phi_i = \mathbb{E}\left[\ell(f_i(X_{D \setminus \{i\}}), Y)\right] - \mathbb{E}\left[\ell(f(X), Y)\right]$ <br> $\phi_0 = \min_{\hat{y}} \mathbb{E}\left[\ell(\hat{y}, Y)\right] - \mathbb{E}\left[\ell(f(X), Y)\right] - \sum_{i \in D} \phi_i$ |
| | $v_f$ | Permutation Test | $\phi_i = \mathbb{E}_{XY}\left[\mathbb{E}_{X_i}\left[\ell(f(X_i, X_{D \setminus \{i\}}), Y)\right]\right]$ <br> $\quad - \mathbb{E}\left[\ell(f(X), Y)\right]$ <br> $\phi_0 = \min_{\hat{y}} \mathbb{E}\left[\ell(\hat{y}, Y)\right] - \mathbb{E}\left[\ell(f(X), Y)\right] - \sum_{i \in D} \phi_i$ |
| | | Conditional Permutation Test | $\phi_i = \mathbb{E}_{XY}\left[\mathbb{E}_{X_i \mid X_{D \setminus \{i\}}}\left[\ell(f(X_i, X_{D \setminus \{i\}}), Y)\right]\right]$ <br> $\quad - \mathbb{E}\left[\ell(f(X), Y)\right]$ <br> $\phi_0 = \min_{\hat{y}} \mathbb{E}\left[\ell(\hat{y}, Y)\right] - \mathbb{E}\left[\ell(f(X), Y)\right] - \sum_{i \in D} \phi_i$ |
| | | Mean Importance | $\phi_i = \mathbb{E}\left[\ell(f(\mathbb{E}[X_i], X_{D \setminus \{i\}}), Y)\right] - \mathbb{E}\left[\ell(f(X), Y)\right]$ <br> $\phi_0 = \min_{\hat{y}} \mathbb{E}\left[\ell(\hat{y}, Y)\right] - \mathbb{E}\left[\ell(f(X), Y)\right] - \sum_{i \in D} \phi_i$ |
| $\{\varnothing\} \cup \{\{i\} \mid i \in D\}$ | $v$ | Univariate Predictors | $\phi_i = \min_{\hat{y}} \mathbb{E}\left[\ell(\hat{y}, Y)\right] - \mathbb{E}\left[\ell(f_i(X_i), Y)\right]$ <br> $\phi_0 = 0$ |
| | | Squared Correlation | $\phi_i = \mathrm{Corr}(X_i, Y_i)^2$ <br> $\phi_0 = 0$ |
| $\mathcal{P}(D)$ | $v$ | Shapley Net Effects | $\phi_i = \phi_i(v)$     (Shapley value) <br> $\phi_0 = 0$ |
| | $v_f$ | SAGE | $\phi_i = \phi_i(v_f)$     (Shapley value) <br> $\phi_0 = 0$ |

Table 2: Summary of datasets. *Split size*: number of examples used for training, validation and testing. *Classes*: the number of possible classes (where applicable).

| Dataset | Features | Examples | Split Size | Classes | Loss | Model Type |
|---|---|---|---|---|---|---|
| Bank Marketing | 16 | 45,211 | (36,169, 4,521, 4,521) | 2 | Cross Entropy | CatBoost |
| Bike Rental | 12 | 10,886 | (8,710, 1,088, 1,088) | – | MSE | XGBoost |
| Credit Default | 20 | 1,000 | (800, 100, 100) | 2 | Cross Entropy | CatBoost |
| Breast Cancer | 50 | 556 | (334, 111, 111) | 4 | Cross Entropy | Logistic Regression |
| MNIST | 784 | 70,000 | (54,000, 6,000, 10,000) | 10 | Cross Entropy | MLP (256, 256) |

When calculating feature importance, our sampling approximation for SAGE (Algorithm 1) was run using draws from the marginal distribution. We used a fixed set of 512 background samples for the bank, bike and credit datasets, 128 for MNIST, and all 334 training examples for BRCA.

When generating random subsets for the correlation experiment in the main text, we first sampled the number of included features $k$ uniformly at random and then generated a random subset of $k$ indices. We used 5,000 random subsets for the bank, bike, credit and BRCA datasets, and 1,500 subsets for MNIST.

## H.2 Additional Results

Global explanations for several datasets are provided for visual comparison. Figure 1 shows the bank marketing explanations, Figure 2 shows the bike demand explanations, and Figure 3 shows the credit

Figure 1: Feature importance for bank marketing dataset.

Figure 2: Feature importance for bike demand dataset.

quality explanations. The error bars in the SAGE explanations indicate 95% confidence intervals. The explanations have clear differences, but the differences are less pronounced than with the higher dimensional datasets (MNIST and BRCA).

To contextualize the computational cost of calculating SAGE values, we performed a direct comparison between our proposed approach and a naive calculation via SHAP. Recall that SAGE values are the expectation of loss SHAP values across the entire dataset, which must be calculated separately for each instance (see Section 3.1 of the main text). We estimated SAGE values using both approaches to compare how quickly each converges.

To quantify SAGE's convergence speed, we ran the sampling algorithm for a variable number of iterations and then calculated the result's mean similarity with the correct values, which were determined by running the sampling algorithm until the convergence criterion was satisfied for $t = 0.01$ (see Section 3.3 of the main text). For SHAP, we determined the mean number of iterations required for a local explanation to converge (again, with $t = 0.01$) and generated multiple local explanations with this number of samples; we then averaged a variable number of local explanations and calculated the similarity with the true SAGE explanation.

Figure 4 shows the results, using both MSE and correlation as metrics. The results show that both approaches eventually converge to the correct values, but SHAP requires 2-4 orders of magnitude more model evaluations. The results could be made more favorable to SHAP by using a more liberal convergence criterion for the local explanations; SAGE shows that spreading computation across examples yields the correct result more efficiently. However, these barely-converged SHAP values would not be useful as local explanations. We conclude that our sampling approach is far more practical when users only require the global explanation.

Figure 3: Feature importance for credit quality dataset.

Figure 4: Convergence comparison between SAGE and an indirect calculation via local SHAP values.