[Reviews · NeurIPS 2020]

Review 1

Summary and Contributions: The authors propose a feature importance method, Shapley additive global importance (SAGE), by summarizing additive importance measures using Shapley values.

Strengths: The use of Shapley values to summarize global feature importance is interesting. Results seem sound and correct, and the problem has high relevance in ML.

Weaknesses: The main limitation of the proposed method is lack of novelty. The use of Shapley values has been tried before, such as in SHAP, so the main contribution here seems to be the approximation of SAGE using sampling. The paper should have then focused more on this particular contribution and on how this could be better than prior related methods. Still on this note, the paper focuses too much on conceptual reviews, but pushes most of the results and discussion to supplements. It is not clear what is the real advantage of SAGE. Computing Shapley values directly is computationally prohibitive even for a reasonable amount of features, so the authors resort to sampling. What should be a good sample size to achieve consistent Shapley values? In the empirical study, for instance, it is hard to really assess SAGE for this same reason: since SAGE scores are random, the authors should have at least included error bars in the results, say, in Figure 1. In any case, the proposed method doesn't seem to perform significantly better than permutation tests and SHAP.

Correctness: The results seem to be correct.

Clarity: The paper is well written and clear.

Relation to Prior Work: Adequate.

Reproducibility: Yes

Additional Feedback:


Review 2

Summary and Contributions: The paper proposes SAGE, a new model-agnostic method for feature importance which takes into consideration feature interactions (usually ignored by other methods). SAGE is building on concepts from prior work, namely the SHAP method, but extending it to global (contrary to the local) explanations for any model. Additionally, the implementation of the proposed algorithm has efficiency advantages over prior work.

Strengths: 1. The method shows to be a solid new tool for feature selection which is of great significance for proper and effective usage of ML in any domain. 2. Being model-agnostic makes the method widely applicable. 3. The writing is very clear and well organized. 4. Overview and discussion of related work is nicely done. 5. Extensive experiments to support the main claims are provided.

Weaknesses: Some questions I have regarding the paper: 1. Is it possible to provide uncertainty estimates for the estimated feature importance? 2. Since it is difficult to assess the feature importance wrt real data sets, I was wondering if the authors thought about simulation studies with synthetic data where the ground truth general model will be available? 3. What would be the implication of using conditional quantile in Eq. 2 instead of conditional mean? Will this make the method more robust to nonsimetric, heteroscedastisic distributions and will this comply with the required properties in 3.1? 4. Can this method be useful/aplicable as regularizer over latent space features, embeddings in deep models? Or the computational overhead will make it inefficient? 5. Is there an experiment where the method is applied on the same dataset only with different models underlaying SAGE? It will be interesting to see if the selected features overlap across different implementations.

Correctness: Yes.

Clarity: Yes, I find the paper very well written. Couple of suggestions: - The presentation can be improved by including a graphical motivating example in the introduction (something on the line of intuition provided on line 125). - In section 2.3 the paragraphs for the 3 different subgroups can be titeled (just bold text the begining), this might improve readability. - I think $m$ was not defined/mentioned prior to Thm 1.

Relation to Prior Work: Yes, I appreciate Table 1 which summarizes prior work. Missing references that could be discussed: - Stefan Depeweg, José Miguel Hernández-Lobato, Steffen Udluft, and Thomas A. Runkler. Sensitivity analysis for predictive uncertainty. In ESANN, 2017. - Schwab, Patrick, and Walter Karlen. "CXPlain: Causal explanations for model interpretation under uncertainty." Advances in Neural Information Processing Systems. 2019.

Reproducibility: Yes

Additional Feedback:


Review 3

Summary and Contributions: In order to understand the inner working of the machine learning models, the paper assesses the role of individual input features in a global sentence. (1)This paper proposes a new feature importance method, named SAGE. SAGE can apply Shapley value to represent the predictive power of subsets of features. (2)This work also introduces a framework of additive importance measures. The framework unifies many existing methods. (3) An efficient sampling-based approximation method is proposed. The method is faster than the naive calulation.

Strengths: (1) Proposing a new feature importance method, Shapley additive global importance (SAGE) which is model-agnostic. (2) Evaluating SAGT on eight datasets, and the quantitative metrics show SAGE can achieve more representative of the predictive power associated with each feature.

Weaknesses: Some important claims are not clearly described, such as: (1) why the method SAGT is model agnostic. (2) How the unifying framework of additive importance measures is used in machine learning methods. Some references are missed, eg. On the 145th line of the paper, "we apply a game-theoretic solution". I can not know the reference of the game-theoretic solution. On the 184th line of the paper, "the mode $f$ is optimal", why it is optimal. I can not find a describition.

Correctness: Some claims need to be described in detail.

Clarity: A good written.

Relation to Prior Work: This work is clearly discussed how this work differs from previous contribution.

Reproducibility: Yes

Additional Feedback:


Review 4

Summary and Contributions: The paper introduces a new method for computing feature importance based on shapley values. The two points that the paper emphasizes are additivity and feature interactions.

Strengths: The paper is theoretically rigorous and provides consistency and convergence theorems for an approximation algorithm for scaling.

Weaknesses: The experiments do not demonstrate any practical value. First of all the datasets are really trivial. The ML algorithms used (MLP, SVM, Logistic Regression) are either outdated or not used in the right context. For example, logistic regression is used in high dimensional datasets. The authors are missing the most popular method that is used for tabular data, such as boosted trees (XGBoost, CatBoost, LightGBM). It would also be absolutely necessary to use datasets from Kaggle competitions. More importantly, they need to replicate winning solutions that do heavy feature engineering with aggregations and conditionals.

Correctness: The theoretical claims are correct, but the empirical results are insignificant.

Clarity: The paper is well written

Relation to Prior Work: The related work is properly analyzed and compared to other approaches. The authors provide a very informative table summarizing the comparison with other methods

Reproducibility: No

Additional Feedback:

[Author Response · NeurIPS 2020]

We thank all the reviewers for their feedback. We are glad to see that the reviewers recognize the relevance of this work
(R1, R2) and that they appreciate the clarity of our framework (R2, R4), the theoretical rigor of our approach (R4)
and the extensiveness of our experiments (R2, R3). We hope this rebuttal addresses your main concerns, and we will
incorporate all revisions into the updated version of this work.

**Novelty.** There are two main contributions in this paper: 1) A theoretical contribution that enhances our understanding
of the trade-offs made by various methods and that shows why Shapley values are well justified for global feature
importance. 2) An algorithmic contribution that shows how focusing on global importance enables us to improve
computational performance by orders of magnitude compared to state-of-the-art local explanation methods.

**Contribution 1: Theory.** One of our main contributions is a new theoretical perspective on global feature importance
that unites several existing methods (R1): we show that they all make different trade-offs regarding the choice of how
to summarize interactions among each feature's contribution to the total predictive power. Revealing this connection
allows explicit reasoning about what were previously implicit trade-offs people made when choosing a particular
method. To address R1's concerns we will attempt to separate this contribution from the broader conceptual review.

**Contribution 2: Computational performance.** While our theory contribution strongly motivates Shapley values
for global feature importance, Shapley values are known to be expensive to compute. Our second main contribution
is an efficient sampling-based estimation algorithm that specifically targets global feature importance. While local
Shapley based sampling methods exist, they are orders of magnitude slower than SAGE when applied to global feature
importance. We apologize if this was obscured in Figure 1, where we showed the convergence of SAGE for a whole
dataset against SHAP for a single instance. In reality, to compute SAGE with SHAP you would need to run it over
hundreds or thousands of instances and then average the results (R1). In the updated Figure 1 we will show this more
direct comparison (where SHAP is much slower).

**Uncertainty and convergence.** R1 and R2 asked about uncertainty estimates—Theorem 2 suggests how to calculate
confidence intervals, and returning these is now default behavior. R1 asked how to determine the number of samples
required—Supplement F describes how to determine convergence automatically using the width of the confidence
intervals, which is now the default mode. Thank you for raising these valuable points.

**Practical value and metrics.** It is widely recognized that measuring feature importance is useful for understanding
which features contribute the most information, for generating scientific hypotheses (see our BRCA experiment), and
for performing feature selection and feature engineering (see R2). Our experiments demonstrate each of these use cases
(R4). Since the first two are harder to evaluate for real datasets (see R2), we focused on unambiguous quantitative
metrics. Our *cumulative importance correlation* metric (Figure 1 middle left) is itself a valuable contribution since this
field lacks reliable quantitative metrics, and it is the most important evaluation for this problem. By this measure SAGE
clearly performs the best overall (see supplementary Table 3). After updating our experiments per R4's request (see
below), SAGE wins on 7/8 datasets (R1). Feature ablation beats SAGE on the Wine dataset but performs far worse on
the remaining ones.

**Models and datasets.** Both R2 and R3 appreciated the extensiveness of our experiments, while R4 mentioned some
concerns. Our goal was to demonstrate the efficiency of our method and the advantages of applying a game-theoretic
approach across a variety of model types and datasets. Using state-of-the-art models, particularly those with heavy
feature engineering, is largely orthogonal to our purpose. In response to R4 we should also point out that the Bike
Sharing dataset was a Kaggle competition and that we used a GBM for our Credit dataset.

Following R4's recommendation, we re-ran experiments with the Bike Sharing, Bank and Credit datasets using XGBoost,
and most methods' importance scores achieved higher correlation with the predictive power. SAGE's advantage over
other methods either stayed the same or improved—on the Bank dataset SAGE now outperforms feature ablation by a
large margin (average correlation of 0.984 vs. 0.921). Since SAGE outperforms other methods on DNNs, GBMs, SVMs,
and random forests individually, we expect SAGE would also outperform other methods on the complex ensembled and
stacked model compilations that typically win Kaggle competitions.

**Clarifications and questions.** The game-theoretic solution is the Shapley value [24] (R3). SAGE is model-agnostic
because it works with any model class, unlike some methods listed in Table 1 (R3). The Bayes classifier minimizes the
population risk when the loss function is cross entropy; this is widely known, but see our Supplement C.1 for a proof
(R3). R2 raised several good questions. Working with simulated datasets is a good suggestion; we considered this but
decided it was worth focusing on data with realistic feature interactions. Using a quantile is an interesting suggestion
that may have certain advantages, but it would remove the connection with mutual information and change properties 1
and 5 from Section 3.1. We agree that applying SAGE as a regularizer would be a valuable application, but it would be
difficult to use out-of-the-box with gradient-based optimization. We also thank R2 for the helpful writing suggestions.

[Meta-Review · NeurIPS 2020]

The reviews were split on the paper with different reviewers focusing on different aspects. There were some concerns about the novelty and the empirical demonstration. However, at the same time, there was also an appreciation for conceptual contributions and the depth (not breadth) of the experiments. Overall, there seems to be a significant contribution made. The authors should reconsider the decision to put most experiments (all but one) in the supplementary material, which made judging the empirical contribution of the paper difficult. But the experimental analysis is well done and shows some convincing evidence. It would be advised to increase the breadth of experiments included in the main paper.